# Gastrointestinal Safety Assessment of GLP-1 Receptor Agonists in the US: A Real-World Adverse Events Analysis from the FAERS Database

**DOI:** 10.3390/diagnostics14242829

**Published:** 2024-12-16

**Authors:** Samuel Prince Osei, Edwin Akomaning, Teodora Francesca Florut, Mohit Sodhi, Brian E. Lacy, Wafa A. Aldhaleei, Akshaya Srikanth Bhagavathula

**Affiliations:** 1Department of Public Health, North Dakota State University, Fargo, ND 58108, USA; samuel.p.osei@ndsu.edu (S.P.O.); edwin.akomaning@ndsu.edu (E.A.); 2Faculty of Health Sciences, Simon Fraser University, Burnaby, BC V5A 1S6, Canada; francescaflorut@gmail.com; 3Faculty of Medicine, University of British Columbia, Vancouver, BC V6T 1Z1, Canada; msodhi@student.ubc.ca; 4Division of Gastroenterology and Hepatology, Mayo Clinic, Jacksonville, FL 32224, USA; lacy.brian@mayo.edu; 5Division of Gastroenterology and Hepatology, Mayo Clinic, Rochester, MN 55905, USA; aldhaleei.wafa@mayo.edu

**Keywords:** glucagon-like peptide-1 receptor, diabetes mellitus, adverse drug reaction reporting systems, gastrointestinal diseases, pharmacovigilance, weight loss agents

## Abstract

**Background:** Glucagon-like peptide-1 receptor agonists (GLP-1 RAs) are commonly used to treat obesity and diabetes but are linked to a variety of gastrointestinal (GI) adverse events (AEs). Real-world data on GLP-1 RA-related GI AEs and outcomes are limited. This study assessed GI AEs and adverse outcomes using the US FDA Adverse Event Reporting System (FAERS). **Methods:** This retrospective pharmacovigilance study used the US FDA FAERS database (2007–2023). We searched GLP-1 RA medications, AEs, and adverse outcomes. Demographic, treatment indication, and AE data were collected. Descriptive analysis involved frequencies and percentages, while reporting odds ratio (ROR), proportional reporting ratio, Bayesian confidence propagation neural network, and multivariate logistic regression were used to analyze GLP-1 RA-related GI AEs and outcomes. **Results:** From 2007 to 2023, a total of 187,757 AEs were reported with GLP-1 RAs, and 16,568 were GLP-1 RA-associated GI AEs in the US. Semaglutide was linked to higher odds of nausea (IC_025_: 0.151, β_Coeff_: 0.314), vomiting (IC_025_: 0.334, β_Coeff_: 0.495), and delayed gastric emptying (IC_025_: 0.342, β_Coeff_: 0.453). Exenatide was associated with pancreatitis (IC_025_: 0.601, β_Coeff_: 0.851) and death (ROR: 4.50, IC_025_: 1.101). Overall, semaglutide had a broader range of notable adverse effects; by comparison, dulaglutide and liraglutide use was associated with fewer significant GI AEs. **Conclusions:** Analysis of the FAERS data reveals that GLP-1 RAs, particularly semaglutide and exenatide, are significantly associated with specific GI AEs, such as nausea, vomiting, delayed gastric emptying, and pancreatitis. Clinicians should be aware of these potential risks to ensure optimal monitoring and patient safety. This study demonstrated the utility of pharmacovigilance data in identifying safety signals, which can inform future pharmacoepidemiological investigations to confirm causal relationships. Clinicians should be aware of these potential risks to ensure optimal monitoring and patient safety.

## 1. Introduction

Glucagon-like peptide-1 receptor agonists (GLP-1 RAs) are injectable antihyperglycemic medications known for effective glycemic control in type 2 diabetes mellitus, reducing glucose levels via insulin secretion and glucagon suppression [1]. They also delay gastric emptying, enhancing their glucose-lowering effects [2]. Additionally, GLP-1 RAs aid weight loss and have been approved as weight loss medications by the United States Food and Drug Administration (FDA) [3]. It has been reported that 54.3 million prescriptions of GLP-1 RAs in the US were submitted between April 2019 and October 2022, with once-weekly injectable dulaglutide and semaglutide being the most common [4].

Despite their proven benefits, GLP-1 RAs are frequently associated with gastrointestinal (GI) adverse events (AEs), leading the FDA to issue safety warnings [5,6,7]. A black box warning was issued in December 2017 advising healthcare providers not to use these agents in patients with a personal or family history of multiple endocrine neoplasia type 2 (MEN 2) [8,9,10,11], which may be associated with a variety of GI symptoms, such as constipation, diarrhea, abdominal pain or cramps, flatulence, and dysphagia [12]. In September 2023, the FDA further highlighted the risk of ileus, intestinal obstruction, and blockage in a revised and updated Ozempic label [13]. GI AEs may range from bothersome symptoms, including nausea, vomiting, abdominal pain, diarrhea, and constipation, to more serious events, such as GI bleeding, gastroparesis, pancreatitis, intestinal obstruction, and cholelithiasis [6,14,15].

The prevalence of GI AEs can significantly affect patients’ quality of life and treatment adherence. However, conflicting evidence exists regarding how frequently AEs occur. One study found that hospitalization, disability, and congenital anomalies accounted for the majority of serious outcomes (13.78%), while death and life-threatening outcomes were much less frequently reported, at 0.75% and 0.90%, respectively [5]. A systematic review and meta-analysis of clinical trials reported benefits in mortality, cardiovascular, and renal outcomes with GLP-1 RAs, highlighting the many advantages of these widely used agents [16]. 

This study aimed to fill this knowledge gap by comprehensively analyzing GLP-1 RA-associated GI AEs and outcomes using the FDA Adverse Events Reporting System (FAERS). The goal of this study was to provide real-world data into the frequency and severity of these adverse events, thereby enabling clinicians to better tailor monitoring and treatment strategies and thus potentially improve patient outcomes. The FAERS database serves as a cornerstone of pharmacovigilance by facilitating the generation of safety reports for suspected drug reactions, which require validation through rigorous pharmacoepidemiological studies.

## 2. Methods

### 2.1. Study Design and Data Source

This is a retrospective pharmacovigilance study conducted using the US FDA’s FAERS database, a post-marketing safety surveillance repository that captures spontaneous reports of AEs, medication errors, and product quality issues. These reports are voluntarily submitted by healthcare professionals, pharmaceutical companies, legal entities, and consumers from the US and internationally. This publicly accessible database provides a robust foundation for evaluating the real-world GI safety profile of the GLP-1 RAs through an analysis of spontaneous reports collected following market approvals.

### 2.2. Data Collection

We accessed all available data from inception (2007) to 2023 (31 December 2023) utilizing search terms related to GLP-1 RA drugs, including “exenatide”, “liraglutide”, “dulaglutide”, “albiglutide”, “lixisenatide”, and “semaglutide”. The descriptions and classifications of AE reports were based on the Preferred Terms (PTs) and 27 System Organ Classes (SOCs) concentrated in the Medical Dictionary for Regulatory Activities (MedDRA) terminology set (version 26.1) released by the International Conference on Harmonization of Technical Requirements for Registration of Pharmaceuticals for Human Use. The generic and brand names of GLP-1 RAs approved by the US FDA were used to identify AEs and outcomes in the DRUG files, including exenatide (BYETTA, BYDUREON), liraglutide (VICTOZA, SAXENDA), dulaglutide (TRULICITY), lixisenatide (ADLYXIN, SOLIQUA), and semaglutide (OZEMPIC, RYBELSUS, WEGOVY). Of note, albiglutide was discontinued from the US market as of July 2017. The summary of approved dates for the included GLP-1 RAs is presented in Appendix A.

### 2.3. Data Extraction and Variables

For this study, GI-related AEs were defined by over 900 PTs in the MedDRA, including some of the most common AEs (e.g., nausea and vomiting) associated with GI system disorders (SOC: 10019212) for GLP-1 RAs. Individual cases of GLP-1 RAs related to GI AEs and outcomes were reviewed and identified by these PTs. The World Health Organization defines a safety signal as “reported information on a possible causal relationship between an AE and a drug, of which the relationship is unknown or incompletely documented previously” [17]. A list of top safety signal strength of GLP-1 RA-associated GI AEs and adverse outcomes ranked by the number of incidence cases at the PT level in the FAERS database is shown in Appendix A. The role codes are identifiers used in pharmacovigilance and clinical research to classify the involvement of medication or substance in an AE [18], with the codes for AEs assigned based on reports to FAERS as primary suspect, secondary suspect, concomitant, or interacting medications. To ensure that the GI AEs were most likely caused by GLP-1 RAs during drug use, the analysis focused on reports where GLP-1 RAs were the primary suspect drug, and cases involving the combination of GLP-1 RAs with other medications were omitted.

Demographics (e.g., sex, age), reporting characteristics (e.g., year, occupation of reporters), indication for the use of GLP-1 RAs (e.g., diabetes mellitus, weight loss, obesity), GI AEs (e.g., nausea, vomiting, abdominal pain, abdominal discomfort, diarrhea, pancreatitis, constipation, delayed gastric emptying, dyspepsia, flatulence, gastroesophageal reflux disease (GERD), gastritis, GI hemorrhage, intestinal obstruction, cholecystitis, peptic ulcer, inflammatory bowel disease, and anal fissure), and adverse outcomes were analyzed. The National Cancer Institute Common Terminology Criteria for Adverse Events (version 5.0) were used to classify serious adverse outcomes as follows: disability and hospitalization [grade 3]; life-threatening conditions [grade 4]; and death [grade 5] [19]. Reports with data errors and missing data were excluded.

### 2.4. Data Mining and Statistical Analysis

Categorical variables were reported as frequencies and percentages for the descriptive evaluation of characteristics and outcome measures of these GI AE reports regarding GLP-1 RAs. Although no gold standard method for signal detection exists, the study employed a multi-method approach for safety signal mining, including the reporting odds ratio (ROR), proportional reporting ratio (PRR), Bayesian confidence propagation neural network (BCPNN), and multivariate logistic regression (MLR) adjusted for confounders, to detect signals of strong associations. The formulae and signal detection criteria of the four disproportionality algorithms are presented in Appendix A. The analysis was performed using Microsoft Excel 2019, SPSS software version 28.0 (IBM Corp., Armonk, NY, USA), and R software version 4.3.3 for statistical analysis. The figures were created using the “ggplot2” package for R language.

### 2.5. Quality Control

Duplicate records and reports with data errors or missing data were excluded to minimize bias and ensure the integrity of the risk signal identification process.

## 3. Results

From 2007 to 2023, a total of 187,757 AEs associated with GLP-1 RAs were recorded. After excluding non-GI AEs (*n* = 126,923) and those related to lixisenatide (*n* = 21) and albiglutide (*n* = 5), 60,808 GI AEs were identified. Subsequent exclusion of reports from outside the US and those with missing data resulted in 16,568 GI AEs being analyzed. The distribution of these AEs among different GLP-1 RAs was as follows: exenatide (26.6%), liraglutide (24.9%), dulaglutide (24.6%), and semaglutide (23.9%). Reports related to albiglutide and lixisenatide use were excluded from the study because they were limited in number. Further details are provided in Figure 1.

The characteristics of GI AEs associated with different GLP-1 RAs in the US are detailed in Table 1. The median age was 61 years, with an interquartile range of (52–69) years. The majority of the reports involved female patients (62.5%) and those aged 18–64 years (61.8%). Most reports (65%) came from consumers; the primary indication was the treatment of diabetes mellitus (91.1%). Hospitalization was the most commonly reported adverse outcome (20.3%), followed by life-threatening conditions and death (1.2% each) and disability (0.8%).

Figure 2 illustrates the proportion of GI AEs reported for exenatide, liraglutide, dulaglutide, and semaglutide. The four most common GI AEs were frequently reported with semaglutide use and included nausea (50.3%), vomiting (30.2%), abdominal discomfort/pain (24.4%), and diarrhea (25.7%). The most frequently recorded serious AE, pancreatitis, was highest with exenatide use (26.7%), while reports of delayed gastric emptying were more frequent with semaglutide use (8.2%).

Table 2 provides a comprehensive analysis of GI AEs related to GLP-1 RAs, stratified by patient characteristics, such as age, sex, and primary indications, including diabetes, obesity, and weight loss. Nausea was the symptom predominantly reported by females (48.9%), those under 65 years of age (44.5%), and those treated for diabetes (45.8%). Abdominal discomfort or pain, diarrhea, and vomiting showed variable prevalence across different patient demographics, with specific conditions like pancreatitis being particularly common in males (21.6%), patients under 65 years (20.7%), and those with diabetes (16.8%). Diarrhea as an AE was more frequent in those aged 65 years and above (24.7%), while vomiting was more prevalent in obese patients (27.8%).

Four distinct algorithms were employed to detect the GI AE signals of GLP-1 RAs. The positive signals of GI AEs and adverse outcomes were classified at PT levels. The GI safety signals indicated significant differences across GLP-1 RA medications, as depicted in Figure 3. The use of GLP-1 RAs was associated with lower reporting odds of GI AEs (ROR: 0.56, 95% CI: 0.46, 0.69). In contrast, the odds for adverse outcomes were higher (ROR: 2.63, 95% CI: 1.88, 3.68), especially with exenatide, which showed a higher ROR for severe outcomes (ROR: 3.79, 95% CI: 3.50, 4.11). Dulaglutide (ROR: 1.07, 95% CI: 1.00, 1.15) and semaglutide (ROR: 1.46, 95% CI: 1.36, 1.57) also demonstrated increased odds for GI AEs.

Table 3 provides a summary of the positive signals of GI AEs associated with different GLP-1 RAs. Exenatide was associated with pancreatitis (IC_025_: 0.601, β_Coeff_: 0.851), GI hemorrhage (IC_025_: 0.405, β_Coeff_: 0.623), cholecystitis (ROR: 3.12, PRR: 3.09, IC_025_: 0.814, β_Coeff_: 1.157), life-threatening conditions (ROR: 3.32, PRR: 2.48, IC_025_: 0.809, β_Coeff_: 1.050), and death (ROR: 4.50, PRR: 4.40, IC_025_: 1.101, β_Coeff_: 1.512). Meanwhile, semaglutide was associated with several GI AEs, including nausea (IC_025_: 0.151, β_Coeff_: 0.314), vomiting (IC_025_: 0.334, β_Coeff_: 0.495), abdominal pain (IC_025_: 0.132, β_Coeff_: 0.211), constipation (PRR: 2.08, IC_025_: 0.668, β_Coeff_: 0.751), diarrhea (IC_025_: 0.258, β_Coeff_: 0.328), flatulence (IC_025_: 0.430, β_Coeff_: 0.506), and delayed gastric emptying (IC_025_: 0.342, β_Coeff_: 0.453).

## 4. Discussion

Our results add to the growing body of literature demonstrating that there are a variety of GI AEs associated with the use of GLP-1 RAs, including the four agents, which were the focus of this study. Each medication exhibited a distinct profile of GI complications, affirming a diversity of side effects despite shared therapeutic targets. Our findings underscore the importance of pharmacovigilance systems like FAERS in generating initial safety signals. While these signals provide critical insights into potential risks, they are not definitive proof of causality. Confirmatory pharmacoepidemiological studies are necessary to validate these signals and ensure they translate into actionable clinical practices. The observed differences among these agents emphasize the need for further investigation into the mechanisms driving specific GI AEs in each GLP-1 RA, potentially influencing individualized therapeutic approaches. While our study focuses on the GI adverse effects associated with GLP-1 RAs, it is important to acknowledge the substantial benefits these medications provide in diabetes control and weight management, as highlighted in prior clinical trials and real-world studies.

Exenatide was notably associated with severe GI AEs, such as pancreatitis, GI hemorrhage, gastritis, and cholecystitis. In addition, analysis of ROR, PRR, BCPNN, and MLR demonstrated an increased likelihood signal for all analyzed serious adverse outcomes, including disability, life-threatening disease, hospitalization, and death. These metrics enhance our confidence in identifying genuine safety signals, as each method detects signals differently and reduces the likelihood of false positives. Our results are consistent with a previous meta-analysis investigating the risk of cholecystitis with exenatide use [20]. However, previous studies did not show a significant difference in GI hemorrhage or pancreatitis risks with exenatide use [14,21]. As such, this is the first study demonstrating a potential link between exenatide use and an increased risk of GI hemorrhage. This is an important finding for clinicians, given the large number of prescriptions written for these agents. Gastritis is a known potential AE associated with exenatide use, which may lead to bleeding in some patients, especially if chronic in nature and left untreated [22]. Evaluation of safety data for liraglutide revealed a more moderate AE profile. Although liraglutide was associated with pancreatitis in this study, evidence from the literature is mixed, with some well-designed large pharmacoepidemiologic studies also demonstrating an increased risk [14,23] and others not demonstrating such risk [24]. This inconsistency highlights the need for studies directly comparing liraglutide’s GI AE risks against other GLP-1 RAs using rigorous adjustment for confounders, such as concurrent medications and diabetes-related complications. As our study did not utilize an active comparator or adjust for confounders by indication, our results may be skewed toward demonstrating a signal indicating an increased risk of pancreatitis. Dulaglutide has been previously linked to symptoms of abdominal discomfort and diarrhea, which is consistent with our findings [25]. 

Semaglutide demonstrated the broadest range of GI AEs in our study, including nausea, vomiting, abdominal pain, constipation, diarrhea, flatulence, delayed gastric emptying, and GERD. Its broad spectrum of effects on GI motility may make it less suitable for patients with pre-existing GI conditions. Semaglutide has a significant effect on GI motility and function. Semaglutide binds to GLP-1 receptors in the stomach, slowing gastric motility [26]. This may lead to symptoms of nausea and vomiting [27], but it may also delay gastric emptying, leading to the development of gastroparesis. In addition, delays in gastric emptying may exacerbate GERD symptoms. Constipation, diarrhea, and flatulence are some of the other most common side effects associated with semaglutide [28]. 

Given the variations in GI AE profiles among different GLP-1 RAs, future research should focus on personalized medicine approaches that consider individual patient risk factors and comorbidities when prescribing these medications. The development of AI-driven smart transdermal delivery systems could facilitate personalized treatment by dynamically adjusting or halting medication delivery in response to the detected AEs, such as gastroparesis or hypoglycemia. These systems are capable of halting medication delivery if AEs such as gastroparesis and hypoglycemia are detected. Such technologies would allow clinicians to respond swiftly to emerging AEs, reducing the risk of prolonged exposure. Prospective studies should aim to collect more granular data on the patient history and concurrent medications to better adjust for potential confounders, ensuring that treatment is tailored to minimize risks and maximize therapeutic benefits. 

Our study had many strengths and limitations. The use of national-level US data provides broad generalizability, especially in identifying rare and serious AEs not frequently captured in clinical trials. This is the first and largest study using US national data that enables generalizability. Our study investigated a wide array of GI AEs using real-world data, which can be advantageous in identifying AEs compared to clinical trial data. Additionally, we utilized robust statistics, including ROR, PRR, BCPNN, and MLR, which enhanced our confidence in identifying genuine safety signals. 

However, our analysis is limited by its retrospective nature and a reliance on self-reported data, which could lead to misclassification of outcomes. The FAERS database primarily captures voluntarily reported adverse events, which may result in underreporting of minor side effects and skew the representation toward more serious outcomes or specific patient demographics. We also acknowledge that the FAERS data may contain duplicate or incomplete records. To mitigate this, we employed rigorous data quality control measures, including the removal of duplicate entries and the exclusion of reports with missing or erroneous data, as outlined in the Methods section. 

Furthermore, the FAERS database is designed for signal detection rather than incidence estimation. It is important to note that the FAERS database is not representative of the entire population using GLP-1 RAs, and caution should be exercised in extrapolating incidence data from this dataset. The absence of detailed classification of the adverse outcomes, the lack of adjustment for significant confounders like smoking and hyperlipidemia, and potential confounding by indication, especially in patients with pre-existing diabetes versus those using GLP-1 RAs for weight loss, are notable limitations that future research should aim to address. Future studies should consider matched or randomized designs to better control for these confounders, allowing for a clearer interpretation of causal relationships. Prospective studies with detailed patient profiling and longitudinal follow-up could provide more definitive evidence of causality and help refine the treatment approaches to minimize GI AEs and assess the long-term impact of drug discontinuation due to adverse effects. 

## 5. Conclusions

Our comprehensive evaluation of GI AEs associated with GLP-1 RAs highlights significant medication-specific risks that require careful consideration in clinical practice. Clinicians should consider the GI AE profile when selecting GLP-1 RAs, particularly for patients with pre-existing GI conditions or significant comorbidities. By tailoring treatment strategies to individual risk profiles and enhancing patient monitoring, healthcare providers can mitigate the impact of these AEs and improve patient outcomes.

## Figures and Tables

**Figure 1 diagnostics-14-02829-f001:**
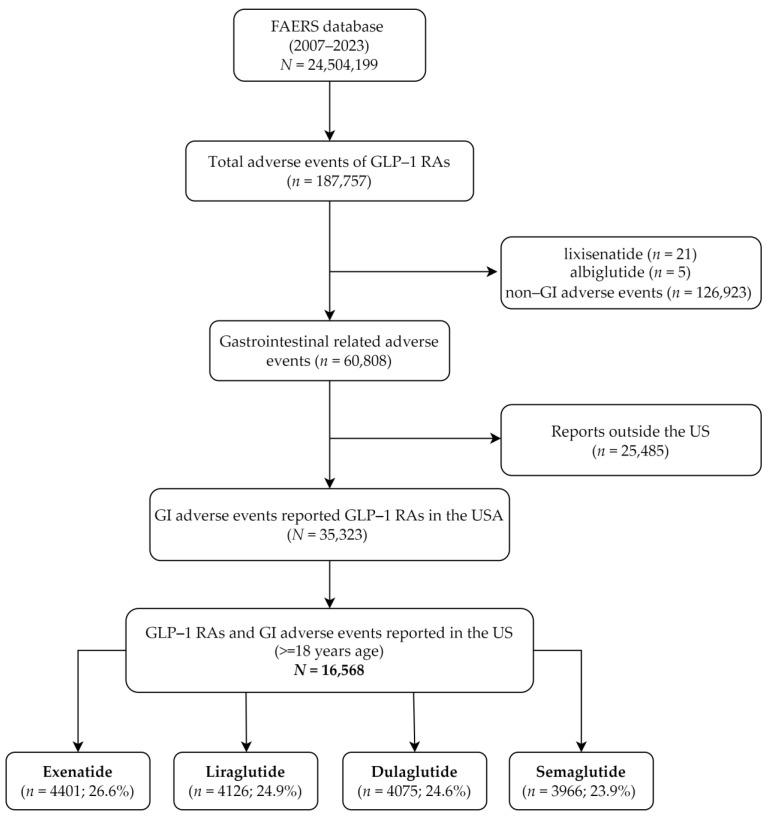
The flow diagram of selection of GLP-1 RA-associated adverse events and outcomes from the FAERS database.

**Figure 2 diagnostics-14-02829-f002:**
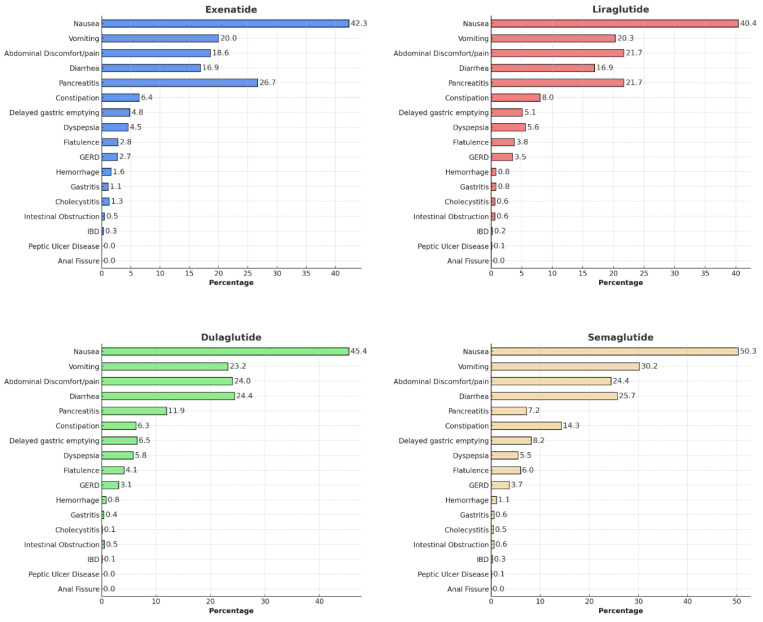
Proportion of gastrointestinal adverse effects reported with different GLP-1 RAs. GERD: gastroesophageal reflux disease; IBD: inflammatory bowel disease.

**Figure 3 diagnostics-14-02829-f003:**
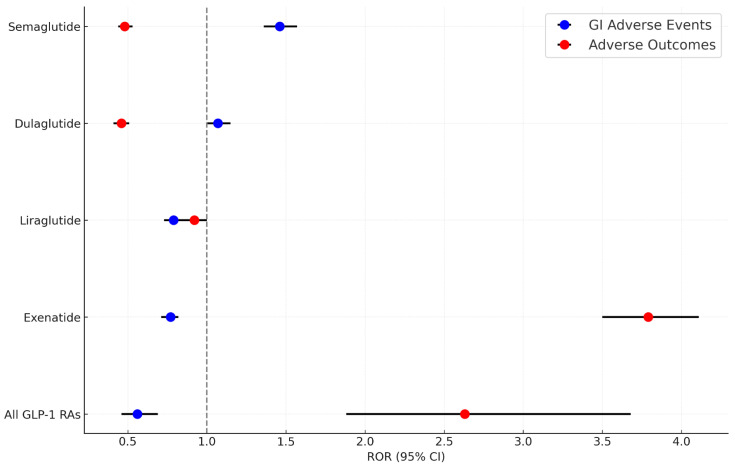
Safety signals of gastrointestinal adverse events and adverse outcomes associated with GLP-1 RAs. ROR: reporting odds ratio; GI: gastrointestinal; GLP-1 RAs: glucagon-like peptide-1 receptor agonists.

**Table 1 diagnostics-14-02829-t001:** Characteristics of GLP-1 RA-associated gastrointestinal adverse reports in the US.

Characteristics, *n* (%)	All GLP-1 RAs (*n* = 16,568)	Exenatide (*n* = 4401; 26.6)	Liraglutide (*n* = 4126; 24.9)	Dulaglutide (*n* = 4075; 24.6)	Semaglutide (*n* = 3966; 23.9)
**Gender**					
Female	10,270 (62.0)	2797 (63.6)	2788 (67.6)	2306 (566)	2379 (60.0)
Male	6152 (37.1)	1591 (36.2)	1325 (32.1)	1684 (41.3)	1552 (39.1)
Not specified	146 (0.9)	13 (0.3)	13 (0.3)	85 (2.1)	35 (0.9)
**Age** ^†^					
18–44	1648 (10)	328 (7.6)	572 (13.9)	385 (9.4)	363 (9.2)
45–54	3262 (19.8)	856 (19.8)	938 (22.7)	857 (21.0)	611 (15.4)
55–64	5282 (32.0)	1579 (36.4)	1320(32.0)	1273 (31.2)	1110 (28.0)
65+	6308 (38.2)	1570 (36.2)	1296 (31.4)	1560 (38.3)	1882 (47.5)
**Reporting Year** ^†^					
2007–2009	1130 (6.8)	1130 (25.7)	-	-	-
2010–2014	2176 (13.1)	1688 (38.4)	486 (11.8)	2 (0)	-
2015–2019	7560 (45.6)	1343 (30.5)	3088 (74.8)	2513 (61.7)	616 (15.5)
2020–2023	5702 (34.4)	240 (5.5)	552 (13.4)	1560 (38.3)	3350 (84.5)
**Reporting Type**					
Consumer	10,756 (65.0)	3166 (72.0)	2458 (59.7)	2672 (65.7)	2460 (62.2)
Healthcare Professional	5785 (35.0)	1231 (28.0)	1660 (40.3)	1398 (34.3)	1496 (37.8)
**Indications** ^†^					
Diabetes Mellitus	11,221 (91.1)	3677 (99.8)	2593 (77.1)	2652 (99.3)	2299 (88.6)
Obesity	370 (3.0)	16 (0.4)	248 (7.4)	9 (0.3)	97 (3.7)
Weight Loss	984 (8.0)	30 (0.8)	609 (18.1)	29 (1.1)	316 (12.2)
**Adverse Outcomes**					
Disability	130 (0.8)	51 (1.2)	24 (0.6)	33 (0.8)	22 (0.6)
Life-Threatening Conditions	195 (1.2)	98 (2.2)	33 (0.8)	42 (1.0)	22 (0.6)
Hospitalization	3365 (20.3)	1593 (36.2)	804 (19.5)	473 (11.6)	495 (12.5)
Death	192 (1.2)	118 (2.7)	36 (0.9)	23 (0.6)	15 (0.4)

^†^ Missing data and may not reach to 100%.

**Table 2 diagnostics-14-02829-t002:** GI adverse events based on patient characteristics ^†^.

GI Adverse Effects	Male	Female	<65	≥65	DM	Obesity	Weight Loss
Nausea	2295 (37.3)	5017 (48.9)	4543 (44.5)	2814 (44.6)	5134 (45.8)	181 (48.9)	529 (53.8)
Abdominal discomfort or pain	1403 (22.8)	2233 (21.7)	2167 (21.2)	1488 (23.6)	2626 (23.4)	70 (18.9)	211 (21.4)
Diarrhea	1365 (22.2)	2071 (20.2)	1893 (18.5)	1560 (24.7)	2470 (22.0)	56 (15.1)	169 (17.2)
Vomiting	1261 (20.5)	2572 (25.0)	2378 (23.3)	1477 (23.4)	2653 (23.6)	103 (27.8)	259 (26.3)
Pancreatitis	1331 (21.6)	1483 (14.4)	2113 (20.7)	726 (11.5)	1888 (16.8)	54 (14.6)	65 (6.6)
Constipation	542 (8.8)	874 (8.5)	715 (7.0)	714 (11.3)	991 (8.8)	42 (11.4)	124 (12.6)
Delayed gastric emptying	366 (5.9)	629 (6.1)	641 (6.3)	371 (5.9)	726 (6.5)	22 (5.9)	75 (7.6)
Dyspepsia	307 (5.0)	573 (5.6)	527 (5.2)	355 (5.6)	650 (5.8)	18 (4.9)	75 (7.6)
Flatulence	284 (4.6)	390 (3.8)	379 (3.7)	303 (4.8)	484 (4.3)	18 (4.9)	44 (4.5)
Gastroesophageal reflux disease	174 (2.8)	353 (3.4)	303 (3.0)	227 (3.6)	385 (3.4)	19 (5.1)	45 (4.6)
Gastritis	40 (0.7)	78 (0.8)	93 (0.9)	26 (0.4)	87 (0.8)	4 (1.1)	7 (0.7)
Hemorrhage	62 (1.0)	115 (1.1)	105 (1.0)	76 (1.2)	131(1.2)	2 (0.5)	8 (0.8)
Intestinal obstruction	45 (0.7)	47 (0.5)	60 (1.7)	33 (0.5)	51 (0.5)	4 (1.1)	6 (0.6)
Cholecystitis	52 (0.8)	54 (0.5)	74 (0.7)	32 (0.5)	68 (0.6)	5 (1.4)	4 (0.4)
Peptic ulcer	2 (0.0)	6 (0.1)	4 (0.0)	3 (0.0)	6 (0.1)	0 (0.0)	0 (0.0)
Inflammatory bowel disease	10 (0.2)	25 (0.2)	27 (0.3)	8 (0.1)	22 (0.2)	2 (0.5)	3 (0.3)
Anal fissure	2 (0.0)	0 (0.0)	2 (0.0)	1 (0.0)	2 (0.0)	0 (0.0)	0 (0.0)

^†^ Missing data were excluded and may not reach 100%. DM: diabetes mellitus; GI: gastrointestinal.

**Table 3 diagnostics-14-02829-t003:** Signal strength of gastrointestinal adverse events of different GLP-1 RAs by the number of incidence cases in the US FAERS database.

GI Adverse Effects	Exenatide					Liraglutide					Dulaglutide					Semaglutide				
	Cases	ROR (95% CI)	PRR (95% CI)	IC025	β−Coeff	Cases	ROR (95% CI)	PRR (95% CI)	IC025	β−Coeff	Cases	ROR (95% CI)	PRR (95% CI)	IC025	β−Coeff	Cases	ROR (95% CI)	PRR (95% CI)	IC025	β−Coeff
Nausea	1862	0.885 (0.826, 0.946)	0.934 (0.877, 0.994)	−0.103	−0.116	1666	0.799 (0.744, 0.858)	0.880 (0.844, 0.918)	−0.173	−0.226	1850	1.037 (0.968, 1.111)	1.020 (0.981, 1.061)	−0.000	0.039	1996	1.361 (1.267, 1.462)	1.179 (1.137, 1.224)	**0.151**	**0.314**
Vomiting	882	0.772 (0.709, 0.840)	0.817 (0.753, 0.888)	−0.271	−0.256	839	0.794 (0.729, 0.866)	0.836 (0.781, 0.895)	−0.252	−0.243	946	0.994 (0.916, 1.078)	0.995 (0.933, 1.061)	−0.056	−0.014	2298	1.612 (1.488, 1.747)	1.427 (1.347, 1.513)	**0.334**	**0.495**
Abdominal discomfort	251	0.799 (0.691, 0.924)	0.811 (0.701, 0.937)	−0.334	−0.225	274	0.991 (0.861, 1.142)	0.992 (0.869, 1.132)	−0.108	0.050	313	1.162 (1.020, 1.324)	1.150 (1.014, 1.304)	**0.111**	**0.211**	269	1.021 (0.886, 1.178)	1.020 (0.893, 1.165)	−0.079	−0.043
Abdominal pain	594	0.751 (0.681, 0.829)	0.785 (0.712, 0.865)	−0.331	−0.286	650	0.956 (0.868, 1.052)	0.963 (0.888, 1.044)	−0.103	−0.045	710	1.090 (0.996, 1.194)	1.075 (0.994, 1.162)	**0.047**	**0.116**	732	1.233 (1.123, 1.354)	1.190 (1.102, 1.286)	**0.132**	**0.211**
Pancreatitis	1174	2.290 (2.104, 2.492)	1.946 (1.792, 2.113)	**0.601**	**0.851**	895	1.493 (1.367, 1.631)	1.386 (1.291, 1.488)	**0.291**	**0.339**	486	0.654 (0.590, 0.725)	0.695 (0.635, 0.762)	−0.601	−0.549	287	0.307 (0.270, 0.349)	0.357 (0.318, 0.401)	−1.351	−1.133
Constipation	281	0.651 (0.569, 0.745)	0.673 (0.588, 0.770)	−0.542	−0.424	329	0.888 (0.781, 1.010)	0.897 (0.797, 1.009)	−0.210	−0.041	257	0.710 (0.619, 0.814)	0.728 (0.639, 0.829)	−0.565	−0.446	568	2.263 (2.022, 2.532)	**2.082** **(1.884, 2.300)**	**0.668**	**0.751**
Diarrhea	744	0.709 (0.648, 0.776)	0.758 (0.694, 0.828)	−0.362	−0.348	995	0.713 (0.651, 0.782)	0.762 (0.707, 0.821)	−0.365	−0.299	697	1.226 (1.131, 1.329)	1.171 (1.097, 1.249)	**0.182**	**0.283**	1020	1.445 (1.329, 1.571)	1.330 (1.248, 1.418)	**0.258**	**0.328**
Hemorrhage	72	1.840 (1.363, 2.483)	1.826 (1.353, 2.464)	**0.405**	**0.623**	34	0.695 (0.478, 1.011)	0.697 (0.481, 1.011)	−0.709	−0.332	32	0.717 (0.491, 1.045)	0.719 (0.491, 1.051)	−0.790	−0.446	43	0.990 (0.702, 1.397)	0.990 (0.704, 1.392)	−0.271	−0.039
Dyspepsia	199	0.794 (0.675, 0.933)	0.803 (0.683, 0.944)	−0.358	−0.232	229	1.057 (0.906, 1.234)	1.054 (0.911, 1.220)	−0.051	0.081	238	1.100 (0.950, 1.275)	1.095 (0.947, 1.265)	**0.025**	0.131	218	1.042 (0.891, 1.220)	1.040 (0.896, 1.207)	−0.069	0.016
Flatulence	123	0.596 (0.489, 0.727)	0.607 (0.498, 0.740)	−0.719	−0.517	157	0.896 (0.747, 1.075)	0.900 (0.756, 1.072)	−0.250	−0.068	167	0.994 (0.836, 1.181)	0.994 (0.838, 1.179)	−0.138	−0.008	236	1.720 (1.463, 2.023)	1.678 (1.439, 1.956)	**0.430**	**0.506**
Delayed gastric emptying	213	0.723 (0.619, 0.844)	0.736 (0.630, 0.859)	−0.454	−0.318	210	0.777 (0.665, 0.909)	0.789 (0.680, 0.914)	−0.382	−0.258	264	1.064 (0.925, 1.224)	1.060 (0.925, 1.214)	−0.017	0.071	326	1.553 (1.355, 1.781)	1.508 (1.328, 1.712)	**0.342**	**0.453**
Gastritis	49	1.946 (1.348, 2.808)	1.935 (1.341, 2.792)	**0.419**	**0.684**	33	1.158 (0.774, 1.733)	1.157 (0.776, 1.726)	−0.134	0.082	15	0.511 (0.298, 0.875)	0.512 (0.299, 0.880)	−1.437	−0.829	22	0.719 (0.452, 1.144)	0.721 (0.454, 1.143)	−0.749	−0.273
GERD	117	0.777 (0.631, 0.957)	0.783 (0.636, 0.964)	−0.425	−0.252	143	1.118 (0.920, 1.359)	1.114 (0.923, 1.345)	−0.022	0.141	125	0.958 (0.786, 1.167)	0.959 (0.787, 1.168)	−0.211	−0.058	145	1.204 (0.991, 1.463)	1.197 (0.992, 1.444)	**0.057**	0.160
Intestinal obstruction	22	0.856 (0.530, 1.383)	0.857 (0.530, 1.384)	−0.531	−0.159	26	1.171 (0.744, 1.845)	1.170 (0.745, 1.838)	−0.158	0.155	22	0.962 (0.603, 1.532)	0.962 (0.597, 1.550)	−0.420	−0.048	23	1.044 (0.651, 1.675)	1.044 (0.653, 1.670)	−0.306	0.049
Cholecystitis	56	**3.123** **(2.130, 4.581)**	**3.096** **(2.111, 4.541)**	**0.814**	**1.157**	26	0.980 (0.629, 1.527)	0.980 (0.631, 1.523)	−0.355	−0.072	5	0.191 (0.078, 0.468)	0.192 (0.078, 0.470)	−3.238	−1.932	19	0.692 (0.421, 1.139)	0.694 (0.423, 1.138)	−0.824	−0.298
Peptic ulcer	0	0.921 (0.186, 4.567)	0.922 (0.186, 4.568)	−1.287	−0.080	3	1.810 (0.432, 7.577)	1.809 (0.433, 7.568)	−0.304	0.628	0	nan	nan	nan	−157.920	3	1.907 (0.456, 7.984)	1.907 (0.456, 7.974)	−0.247	0.628
Anal fissure	1	1.382 (0.125, 15.250)	1.382 (0.125, 15.249)	−1.273	0.663	1	1.508 (0.137, 16.634)	1.508 (0.137, 16.624)	−1.179	0.539	0	nan	nan	nan	−16.912	1	1.589 (0.144, 17.528)	1.589 (0.144, 17.517)	−1.122	0.597
IBD	14	1.846 (0.938, 3.633)	1.843 (0.936, 3.628)	**0.186**	0.614		0.753 (0.329, 1.726)	0.754 (0.330, 1.725)	−0.978	−0.292		0.464 (0.165, 1.307)	0.465 (0.164, 1.316)	−2.028	−0.930		1.272 (0.610, 2.650)	1.271 (0.611, 2.644)	−0.268	0.248
**Adverse Outcomes**																				
Disability	51	1.794 (1.259, 2.555)	1.785 (1.253, 2.542)	**0.350**	**0.587**	24	0.681 (0.437, 1.062)	0.683 (0.439, 1.062)	−0.792	−0.374	33	1.032 (0.703, 1.515)	1.032 (0.696, 1.530)	−0.248	0.011	22	0.645 (0.407, 1.022)	0.647 (0.410, 1.022)	−0.880	−0.428
Life-Threatening Conditions	98	**3.328** **(3.074, 3.603)**	**2.485** **(2.304, 2.681)**	**0.809**	**1.050**	33	0.934 (0.855, 1.020)	0.947 (0.882, 1.016)	−0.114	−0.467	42	0.515 (0.465, 0.571)	0.572 (0.522, 0.626)	−0.886	−0.227	22	0.484 (0.436, 0.536)	0.548 (0.502, 0.599)	−0.779	−0.909
Hospitalization	1593	2.834 (2.136, 3.760)	**2.793** **(2.105, 3.706)**	**0.783**	**1.205**	804	0.611 (0.420, 0.890)	0.614 (0.423, 0.892)	−0.867	−0.094	473	0.874 (0.625, 1.223)	0.876 (0.624, 1.230)	−0.458	−0.829	495	0.401 (0.257, 0.625)	0.404 (0.260, 0.629)	−1.478	−0.711
Death	118	**4.502** **(3.360, 6.033)**	**4.408** **(3.290, 5.907)**	**1.101**	**1.512**	36	0.693 (0.482, 0.998)	0.696 (0.485, 0.998)	−0.703	−0.321	33	0.484 (0.314, 0.747)	0.487 (0.316, 0.752)	−1.420	−0.906	15	0.267 (0.157, 0.452)	0.269 (0.159, 0.456)	−2.101	−1.370

CI: confidence interval; GERD: gastroesophageal reflux disease; IBD: inflammatory bowel disease; IC: information component; IC025—the lower limit of 95% CI of the IC; PRR: proportional reporting ratio; ROR: reporting odds ratio. β coefficient of multivariate logistic regression—adjusted for age and sex. Significant at *p* < 0.05 were bolded.

## Data Availability

The original contributions presented in the study are included in the article. Further inquiries can be directed to the corresponding author.

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
