# Peer review of "Gastrointestinal Safety Assessment of GLP-1 Receptor Agonists in the US: A Real-World Adverse Events Analysis from the FAERS Database"

_diagnostics, 2024, doi:10.3390/diagnostics14242829_

Round 1

Reviewer 1 Report

Comments and Suggestions for Authors

It is a well orchestrated review of the existing data about the use f GLP-1R agonists. The collection of the data is good, the analysis is reasonable and conclusions are valid.

Author Response

Reviewer comment: It is a well orchestrated review of the existing data about the use f GLP-1R agonists. The collection of the data is good, the analysis is reasonable and conclusions are valid.
Authors response: We sincerely thank you for their positive feedback on our manuscript and for recognizing the robustness of our data collection, analysis, and conclusions. 

Reviewer 2 Report

Comments and Suggestions for Authors

This is a timely and relevant topic and the science presented here aims to fill an important gap in our knowledge about real-world side effects of GLP1RAs.

Overall, the data are presented clearly and the analysis is sound.

I do have some concerns about the methodology used- that is, the use of the FAERs dashboard to mine data.  Do the authors have concern about reporting bias?  My understanding of this may be limited but I don't think most physicians prescribing GLP1RAs are reporting side effects (especially minor side effects) to this database.  Therefore, is the data more likely to represent certain adverse effects and/or certain patient populations?    Brief reading about the FAERs database suggests that the data is not validated and may contain many duplicate or incomplete records; furthermore the FDA website cautions about using this data to extrapolate incidence data. 

This being said, I think that the authors are careful not to conflate prevalence or presence/absence of symptoms reported with incidence, and are clear to report Reporting Odds Ratio (and appropriately do not try to extrapolate more broadly).

I am curious about the fact that relatively equal numbers of reports exist for exenatide, liraglutide, dulaglutide and semaglutide.  Does this reflect prescribing practice during this time period?  I am wondering whether there are any differential effects in patients treated for obesity vs. diabetes, whether this is able to be stratified in the current dataset?  If not possible, this does not limit the value of this work. 

Author Response

Comment 1:  This is a timely and relevant topic and the science presented here aims to fill an important gap in our knowledge about real-world side effects of GLP1RAs.

Overall, the data are presented clearly and the analysis is sound.

I do have some concerns about the methodology used- that is, the use of the FAERs dashboard to mine data.  Do the authors have concern about reporting bias?  My understanding of this may be limited but I don't think most physicians prescribing GLP1RAs are reporting side effects (especially minor side effects) to this database.  Therefore, is the data more likely to represent certain adverse effects and/or certain patient populations?    Brief reading about the FAERs database suggests that the data is not validated and may contain many duplicate or incomplete records; furthermore the FDA website cautions about using this data to extrapolate incidence data. 

Authors response:  We thank you for highlighting the importance of our study and for acknowledging the clarity of our data presentation and the soundness of our analysis. We appreciate your thoughtful feedback regarding potential limitations associated with the use of the FAERS database. Agreeing with your comments we have added the following paragraph in our limitation section. 

“The FAERS database is primarily captures voluntary reported adverse events, which may result in underreporting of minor side effects and skew the representation towards more serious outcomes or specific patient demographics. We also acknowledge that the FAERS data may contain duplicate or incomplete records. To mitigate this, we employed rigorous data quality control measures, including the removal of duplicate entries and the exclusion of reports with missing or erroneous data, as outlined in the Methods section. Furthermore, the FAERS database is designed for signal detection rather than incidence estimation. It is important to note that the FAERS database is not representative of the entire population using GLP-1 RAs, and caution should be exercised in extrapolating incidence data from this dataset.”

Comment 2: This being said, I think that the authors are careful not to conflate prevalence or presence/absence of symptoms reported with incidence, and are clear to report Reporting Odds Ratio (and appropriately do not try to extrapolate more broadly).

I am curious about the fact that relatively equal numbers of reports exist for exenatide, liraglutide, dulaglutide and semaglutide.  Does this reflect prescribing practice during this time period?  I am wondering whether there are any differential effects in patients treated for obesity vs. diabetes, whether this is able to be stratified in the current dataset?  If not possible, this does not limit the value of this work. 

Authors response: We appreciate you for acknowledging our careful approach and emphasis on using ROR. We believe the relieve the relative equal number of reports for these four drugs may partially reflect evolving prescribing practices during this period. However, this observation could also be influenced by reporting bias inherent in the FAERS database, as such we discussed in the discussion section.

We appreciate your interest to stratifying AEs by obesity vs. diabetes. This could be done in another research hypothesis due to higher data analysis presented in this paper.

Reviewer 3 Report

Comments and Suggestions for Authors

Very useful article , i hope the authors add data about successful diabetes control, overcoming obesity. It is a new large-scale study.  The conclusion is clear

Also there is no data about drug discontinuation outcomes. I recommend adding data about drug discontinuation and the outcome of the drug even after causing the side effects.

However, the article deserve publication.

Author Response

Comment 1: Very useful article , i hope the authors add data about successful diabetes control, overcoming obesity. It is a new large-scale study.  The conclusion is clear.

Authors response: We highly appreciate your wonderful suggestions. While we agree that the primary focus of this study is on the GI AEs of GLP-1 RAs, we acknowledge the importance of presenting additional context regards their therapeutic benefits. We have included the following statement in the discussion section

" While our study focuses on the GI adverse effects associated with GLP-1 RAs, it is important to acknowledge the substantial benefits these medications provide in diabetes control and weight management, as highlighted in prior clinical trials and real-world studies." 

Comment 2: Also there is no data about drug discontinuation outcomes. I recommend adding data about drug discontinuation and the outcome of the drug even after causing the side effects.

However, the article deserve publication.
Authors response: We agree with your comment, and drug discontinuation is critical for clinical decision-making. Unfortunately, the FAERS database doesn't have follow-up data after discontinuation. To address this limitation, we have added the following statement in the limitation section "Prospective studies with detailed patient profiling and longitudinal follow up could provide more definitive evidence of causality and help refine treatment approaches to minimize GI AEs and assess the long-term impact of drug discontinuation due to adverse effects